⊕ | **Open Peer Review** | Host-Microbial Interactions | Research Article

# *Klebsiella pneumoniae* OmpR facilitates lung infection through transcriptional regulation of key virulence factors

Axel B. Janssen,[1] Vincent de Bakker,[1] Rieza Aprianto,[2] Vincent Trebosc,[3] Christian Kemmer,[3] Michel Pieren,[3] Jan-Willem Veening[1]

**ABSTRACT** Bacteria must adapt to the stresses of specific environmental conditions to survive. This adaptation is often achieved by altering gene expression through two-component regulatory systems (TCSs). In Gram-negative bacteria, the response to environmental changes in osmolarity and pH is primarily mediated by the EnvZ/OmpR TCS. Although the functioning of EnvZ/OmpR has been well characterized in *Escherichia coli*, *Salmonella enterica*, and the *Yersinia* genus, the importance of EnvZ/OmpR TCS in the opportunistic human pathogen *Klebsiella pneumoniae* has been limitedly studied. Here, we investigated the importance of EnvZ/OmpR in *K. pneumoniae* for fitness, gene regulation, virulence, and infection. Through the generation of a markerless *ompR*-deletion mutant, we show that overall fitness of *K. pneumoniae* is not impacted *in vitro*. Using dual RNA sequencing of *K. pneumoniae* co-incubated with human lung epithelial cells, we demonstrate that the *K. pneumoniae* OmpR regulon includes important virulence factors but shows otherwise limited overlap with the regulons of other Gram-negative bacteria. In addition, we show that deletion of *ompR* in *K. pneumoniae* leads to a stronger antibacterial transcriptional response in human lung epithelial cells. Lastly, we show that OmpR is crucial for *K. pneumoniae* virulence and infection through a murine lung infection model. As the adaptation of commensal bacteria to specific niches is mediated by TCSs, we show that EnvZ/OmpR plays a crucial role in successful lung infection, as well as in virulence. These results suggest that OmpR is an interesting target for anti-virulence drug discovery programs.

**IMPORTANCE** Bacteria use two-component regulatory systems (TCSs) to adapt to changes in their environment by changing their gene expression. In this study, we show that the EnvZ/OmpR TCS of the clinically relevant opportunistic pathogen *Klebsiella pneumoniae* plays an important role in successfully establishing lung infection and virulence. In addition, we elucidate the *K. pneumoniae* OmpR regulon within the host. This work suggests that *K. pneumoniae* OmpR might be a promising target for innovative anti-infectives.

**KEYWORDS** *Klebsiella pneumoniae*, OmpR, host-pathogen interaction, epithelial cells, infection, dual RNA-seq

K *lebsiella pneumoniae* is an important nosocomial pathogen due to the rapidly increasing rate of multidrug resistance. The spread of strains resistant to fluoroquinolones, third-generation cephalosporins, aminoglycosides, and increasing spread of resistance to last-line antibiotics such as carbapenems and colistin have limited treatment options (1, 2). This has made *K. pneumoniae* one of the pathogens with the highest burden of antimicrobial resistance deaths (3). Due to this multidrug resistance and the lack of development of novel antibiotics, *K. pneumoniae* has been named by the World Health Organization as a critical priority pathogen for the development of novel antibiotics (4).

Address correspondence to Jan-Willem Veening, Jan-Willem.Veening@unil.ch, or Michel Pieren, Michel.Pieren@bioversys.com.

V.T., C.K., and M.P. own equity in BioVersys AG. The other authors declare no competing interests.

See the funding table on p. 14.

 10.1128/spectrum.03966-23 **1**

*K. pneumoniae* can asymptomatically colonize the upper respiratory tract, skin, and digestive tract in healthy individuals. However, the asymptomatic colonization of these niches by *K. pneumoniae* may subsequently lead to infections such as lung infections, soft tissue infections, wound infections, and urinary tract infections (5). The populations that are particularly at risk of *K. pneumoniae* infections are infants, the elderly, the immunocompromised, and hospitalized patients (5, 6). Persistent asymptomatic niche colonization by potential pathogens is mediated by adaptation to niche-specific conditions. Two-component regulatory systems (TCSs) mediate the adaptation of bacteria in response to changing environmental factors through adaptation of gene expression. TCSs are composed of a membrane-bound sensor histidine kinase and a cytosolic transcriptional response regulator (7, 8). The membrane-bound sensor kinase will activate through *trans*-autophosphorylation in response to a specific environmental signal (8, 9). Next, the phosphorylated sensor histidine kinase will transfer its phosphoryl group to the response regulator, leading to conformational changes within the response regulator. The phosphorylated response regulator will then bind to specific promoter sequences, resulting in changed gene expression and an increased survival under the specific environmental conditions (10–12). TCSs thus play an important role in persistent colonization and subsequent infection, by potential pathogens like *K. pneumoniae*, in diverse niches.

The EnvZ/OmpR TCS is involved in the adaptive response to environmental changes in both osmolarity and pH (13–15). EnvZ is the sensor histidine kinase part of this TCS. In the presence of specific activation signals, the EnvZ dimer will phosphorylate the OmpR transcriptional response regulator. Two phosphorylated OmpR molecules will bind to the consensus target sequences, resulting in an altered gene expression (11, 16). The role of EnvZ/OmpR in gene regulation, virulence, colonization, and infection has been well studied in several bacterial species, like *Escherichia coli* (including uropathogenic and adherent invasive strains) (13, 17–22), *Yersinia* spp. (23–25), *Salmonella enterica* (26, 27), *Shigella flexneri* (28), and *Acinetobacter baumannii* (29), often suggesting that EnvZ/OmpR plays a key role in pathogenesis. The number of studies in *K. pneumoniae* has, however, been limited.

Recently, it has been shown that *K. pneumoniae* ATCC 43816 deficient in OmpR has reduced virulence in a mouse lung infection model (30). This reduction in virulence has been suggested to be the result of the loss of hypermucoviscosity because of the loss of OmpR. Although OmpR seems to be involved in regulation of hypermucoviscosity *in vitro*, the transcriptional changes upon deletion of *ompR* in a host-pathogen setting have not been elucidated. Here, we studied the functioning of the EnvZ/OmpR TCS in *K. pneumoniae* ATCC 43816 on fitness, gene expression, and infection and virulence. We utilized dual RNA sequencing (RNA-seq) (31) to elucidate the role of OmpR as a transcriptional regulator in early host-pathogen interaction between *K. pneumoniae* and human lung epithelial cells. In addition, we confirm the role of OmpR in infection and virulence through a lung infection mouse model.

## MATERIALS AND METHODS

### Bacterial strains, growth conditions, and mutant construction

*K. pneumoniae* ATCC 43816 WT was obtained from the American Type Culture Collection (ATCC; Manassas, VA, USA). *K. pneumoniae* ATCC 43816 WT and derivate strains were cultured in lysogeny broth (LB) without antibiotics, at 37°C with agitation, unless otherwise noted. The deletion of *ompR* was performed through an allelic exchange method using a combination of positive selection (selection through sodium tellurite) and negative selection (selection through SceI restriction endonuclease), as described before (18, 32).

## Determination of maximum specific growth rate

Maximum specific growth rates were determined by measuring growth in a Tecan Infinite M Plex Plate Reader (Tecan, Männedorf, Switzerland). Overnight cultures in LB or RPMI1640 without phenol red (Gibco, Life Technologies, Bleiswijk, The Netherlands) supplemented with 1% fetal bovine serum (FBS; Gibco) were used to inoculate 200 µL of the same medium 1:1,000. Samples were incubated at 37°C with agitation. Growth was observed by measuring the absorbance at 595 nm ($Abs_{595}$) every 10 min. Maximum specific growth speed was determined by calculating the maximum increase in natural logarithm-transformed absorbance values between two adjacent timepoints, divided by time between timepoints in hours: $\mu_{\max} = \max\left(\dfrac{\ln(Abs595_t) - \ln(Abs595_{t-1})}{t - t_{-1}}\right)$.

## Determination of minimal inhibitory concentration

Minimal inhibitory concentrations (MICs) were determined by microbroth dilution method in cation-adjusted Mueller-Hinton broth according to Clinical and Laboratory Standards Institute (CLSI) guidelines, except for the MIC of ertapenem from the ATCC 43816 WT strain that was determined through a Vitek 2, using an AST-GN69 card (BioMérieux, Marcy-l'Étoile, France). The CLSI quality control strain *E. coli* ATCC 25922 was included in the microbroth dilution assays as control.

## *In vitro* infection studies

The human type II lung epithelial cell line A549 (ATCC CCL-185) was cultured in DMEM/F12 with GlutaMAX (Life Technologies) supplemented with 10% FBS (FBS; VWR, Amsterdam, The Netherlands) under controlled conditions [37°C, 5% (vol/vol) $CO_2$]. Prior to the start of the infection experiment, epithelial monolayers were incubated for 10 days after confluence in medium without antibiotics. Confluent monolayers of A549 were co-incubated with either *K. pneumoniae* WT or Δ*ompR* strain at a multiplicity of infection (MOI) of 10. The MOI of 10 was chosen so that a minimum amount of pathogen per host cell was present to provoke an interspecies transcriptional interaction, while minimizing cell death as a result of the infection within the timeframe of exposure. The infection experiment was performed in RPMI1640 medium without phenol red with 1% FBS. A spin infection at refrigerated temperatures was performed ($2,000 \times g$, 5 min, 4°C) to promote the contact between bacterial and epithelial cells immediately after the addition of the bacterial suspension, while minimizing transcriptional changes as the result of the centrifugal forces. After spin infection, the samples were co-incubated at 37°C and 5% $CO_2$ for 2 hours to mimic early infection but to minimize epithelial cell death as a result of the exposure to the bacterial cells. After incubation at 37°C and 5% $CO_2$, the total RNA from both the *K. pneumoniae* and A549 cells was isolated.

## RNA isolation

To minimize transcriptional changes due to sample handling, we did not wash or separate the mixture of bacterial and human cells before RNA isolation. Instead, total RNA was simultaneously isolated from the harvested mixture of co-incubated *K. pneumoniae* and A549 cells. To prevent protein-dependent RNA degradation, we treated the cellular mixture with a saturated buffer solution of ammonium sulfate [pH 5.2, 700 g/L $(NH_4)_2SO_4$], also containing 20 mM EDTA and 25 mM sodium citrate (33). Three parts of saturated solution of ammonium sulfate were mixed to one part of infection medium. The suspension was vigorously pipetted to ensure the complete mixing of the saturated buffer solution and infection medium. After scraping of the adherent host cells, the suspension was incubated further at room temperature for 5 min. The suspension was collected and centrifuged at full speed (20 min, 4°C, $10,000 \times g$). The supernatant was removed, and the cell pellet was snap frozen in liquid nitrogen.

A PCR tube full of sterile, RNase-free 100-µm glass beads (BioSpec, Bartlesville, OK, USA) was mixed with 50 µL 10% SDS and 500 µL phenol-chloroform in a 1.5 mL screw

cap tube. The frozen cell pellet was resuspended in TE buffer (10 mM Tris-HCl, 1 mM Na$_2$DTA, and pH 8.0, in DEPC-treated milliQ H$_2$O). The suspension was added into the screw cap tube and bead beaten in three cycles of 45 seconds each. Tubes were immediately placed on ice and centrifuged at full speed at 4°C to separate organic and aqueous phases. After removing the aqueous phase, back extraction was performed on the organic phase for optimal RNA yield. Phenol was further depleted from the aqueous phase by another round of chloroform extraction. After vigorous vortexing, the mixture was again centrifuged at full speed, 4°C. The aqueous phase was transferred into a fresh eppendorf tube, and nucleic acids were precipitated by the addition of 50 µL 3M NaOAc and 1 mL of cold isopropanol and vigorous mixing. The mixture was incubated for at least 30 min at −20°C before pelleting by centrifugation (full speed, 4°C). Supernatant was carefully removed, and the nucleic acid pellet was washed twice by resuspension in ice-cold 75% ethanol before re-pelleting (full speed, 4°C). The pellet was air dried before DNase treatment.

DNase treatment was performed using recombinant RNase-free DNase I (Roche, Basel, Switzerland) according to the manufacturer's protocols for 1 hour, at room temperature. To remove DNase and gDNA-derived nucleotides, phenol-chloroform extraction, chloroform extraction, isopropanol precipitation, and ethanol washing were performed as previously mentioned. Total RNA was resuspended in 30µL TE buffer. The quantity and quality of total RNA were estimated by NanoDrop, and a 1% bleach gel was employed to assess the presence of genomic DNA and rRNA bands (23S; 2.9 kbp, and 16S; 1.5 kbp) (34).

## Library preparation and sequencing for dual RNA-seq

The quality of the total RNA isolated was checked using chip-based capillary electrophoresis. Samples were simultaneously depleted from human and bacterial ribosomal RNAs by dual-rRNA depletion as previously described (33). Reverse stranded cDNA library preparation was performed with the TruSeq Stranded Total RNA Sample Preparation Kit (Illumina, San Diego, CA, USA) according to the manufacturer's protocol. A single lane of a high-output flowcell in an Illumina NextSeq 500 (Illumina) instrument was used to perform the single-ended sequencing yielding 85-bp reads.

## RNA-seq data analysis

Quality of raw sequencing reads was checked using FastQC (v0.11.9; available at https://www.bioinformatics.babraham.ac.uk/projects/fastqc/; Babraham Bioinformatics, UK). Reads were trimmed for adapter sequences from the TruSeq3-SE library preparation kit, minimal quality of starting and trailing nucleotides (Phred score of 20), a cutoff of an average Phred score of 20 in a 5-nucleotide sliding window, and length (minimum length 50 nucleotides) using Trimmomatic (v0.39) (35). The quality of trimmed reads was confirmed using FastQC.

Quality-filtered reads were aligned to a chimeric sequence created by concatenating the circular genome of *K. pneumoniae* ATCC 43816 (GenBank accession GCA_016071735.1) (36) and the human genome (GenBank accession GCA_009914755.4) (37). The corresponding annotation files were downloaded from the same services. First, indexing of the chimeric genome was performed using RNA-STAR with the following options: --alignIntronMax set to 1 and --sjdbOverhang set to 84. Then, alignment was performed by using RNA-STAR (v2.7.10a) (38) with default settings. The aligned reads were summarized through featureCounts (v2.0.1) (39) according to the chimeric annotation file in reverse stranded (-s 2), multimapping (-M), fractionized (--fraction), and overlapping (-O) modes. The single-pass alignment onto the chimeric genome was selected to minimize the false discovery rate. However, due to this approach, we had to adjust the summarizing process, considering the overlapping nature of bacterial genes and their organization into operon structures. In case of conflicting gene annotations between species, the naming in these files was deemed leading.

The counts of the aligned reads of the host and pathogen libraries were analyzed separately in R (v4.1.1). Differential expression analysis was performed with DESeq2 v1.34.0 (40). For principal component analyses, counts were normalized with a blind rlog transformation. Since coverage for the bacterial data set was high and well saturated, we used stricter parameters than the default for hypothesis testing: $1.5 \times$ depletion or enrichment of mRNAs [$|\log_2FC|>\log_2(1.5)$] with $P_{adj} < 0.05$ as significance threshold. As the host data set was less well saturated, we left the $\log_2FC$ parameter at its default ($|\log_2FC| > 0$: any difference) to look for more subtle effects but restricted the significance threshold to $P_{adj} < 0.01$ to retain high confidence in called hits. Gene Ontology (GO) term enrichment and gene set enrichment analyses were performed with clusterProfiler (v4.2.2) (41) using the $\log_2FC$ values from DESeq2, shrunk with the apeglm (42) method as implemented in that package. GO terms were clustered with the aPEAR package (43).

To assess whether our RNA samples still contained leftover genomic DNA, we inspected the read alignments for reads mapped onto intergenic regions. The presence of these reads indicate either read-through transcription at these areas or the presence of some residual genomic DNA. These reads will, however, not have been counted toward the counts of any individual genes and, as such, are not included in the statistical analysis. The reads coming from residual genomic DNA will have taken up a minor portion of the total sequencing capacity. This is unlikely to affect the statistical power significantly.

## *In vivo* lung infection studies

Male CD-1 mice (15 animals per bacterial strain and inoculum) were infected intranasally with $10^5$ or $10^6$ CFU (50 µL) of exponential phase-cultured *K. pneumoniae* ATCC 43816 WT or $\Delta ompR$. Five animals per group were euthanized using $CO_2$ at 2, 24, and 48 hours post-infection (hpi). Aseptically sampled lungs and blood were assessed for bacterial titers through CFU counting. Animals were sacrificed through inhalation of isoflurane. Terminal blood samples were collected by an intracardiac puncture into K3-EDTA tubes. Blood samples were placed on ice, serially diluted, plated on TSA plates, and incubated overnight at 37°C. The lungs were aseptically removed and placed into 4 mL of sterile saline solution and homogenized using the Cryolys Evolution Precellys System. Serial dilutions of the homogenate were plated on Triptic Soy Agar (TSA) plates and incubated overnight at 37°C. The bacterial counts were determined after overnight incubation.

Survival rate, clinical severity scores, and body weight were recorded during the experimental phase. The scoring of the clinical severity of the disease was as follows: 0: normal; 0.5: mild signs of one of the following symptoms: localized piloerection, hunching, reduced activity, or erratic breathing; 1: one of the following, piloerection, hunched posture, reduced activity, or erratic breathing; 1.5: one of the following piloerection, hunched posture, reduced activity, or erratic breathing PLUS mild signs of one of any of these symptoms; 2: two of the following: piloerection, hunched posture, reduced activity, or erratic breathing; 3: three of the following piloerection, hunched posture, reduced activity, and erratic breathing; 4: moribund; and 5: death.

## Statistical analyses

Except for host differential expression analysis, statistical significance was defined as a *P*-value of <0.05 for all tests. Significance was defined as a $P \leq 0.05$ (*), $P \leq 0.01$ (**), $P \leq 0.001$ (***), or $P \leq 0.0001$ (****). Statistical analyses other than RNA-seq analyses were performed using GraphPad Prism 6 software (GraphPad Software, San Diego, CA, USA) and R (v4.0.3).

## RESULTS

### Deletion of *ompR* does not impact fitness of *K. pneumoniae* ATCC 43816 *in vitro*

As a limited number of studies on the role of EnvZ/OmpR in *K. pneumoniae* have previously been performed, we first aimed to characterize the effects of the deletion of *ompR* in *K. pneumoniae* ATCC 43816 *in vitro*. We constructed a markerless deletion mutant through an allelic exchange method using a combination of positive and negative selection (32). To assess the effects of the deletion of *ompR* on fitness, we determined the growth kinetics of *K. pneumoniae* ATCC 43816 and the Δ*ompR* strain. We observed that the deletion of *ompR* did not lead to an impairment of overall growth kinetics (Fig. 1A). To quantify the growth kinetics in more detail, we determined the maximum specific growth rate, which is an accepted proxy for overall fitness of a bacterial strain (44). Through tracking of the growth, we observed that the maximum specific growth rate of the Δ*ompR* strain was not affected compared with the wild-type

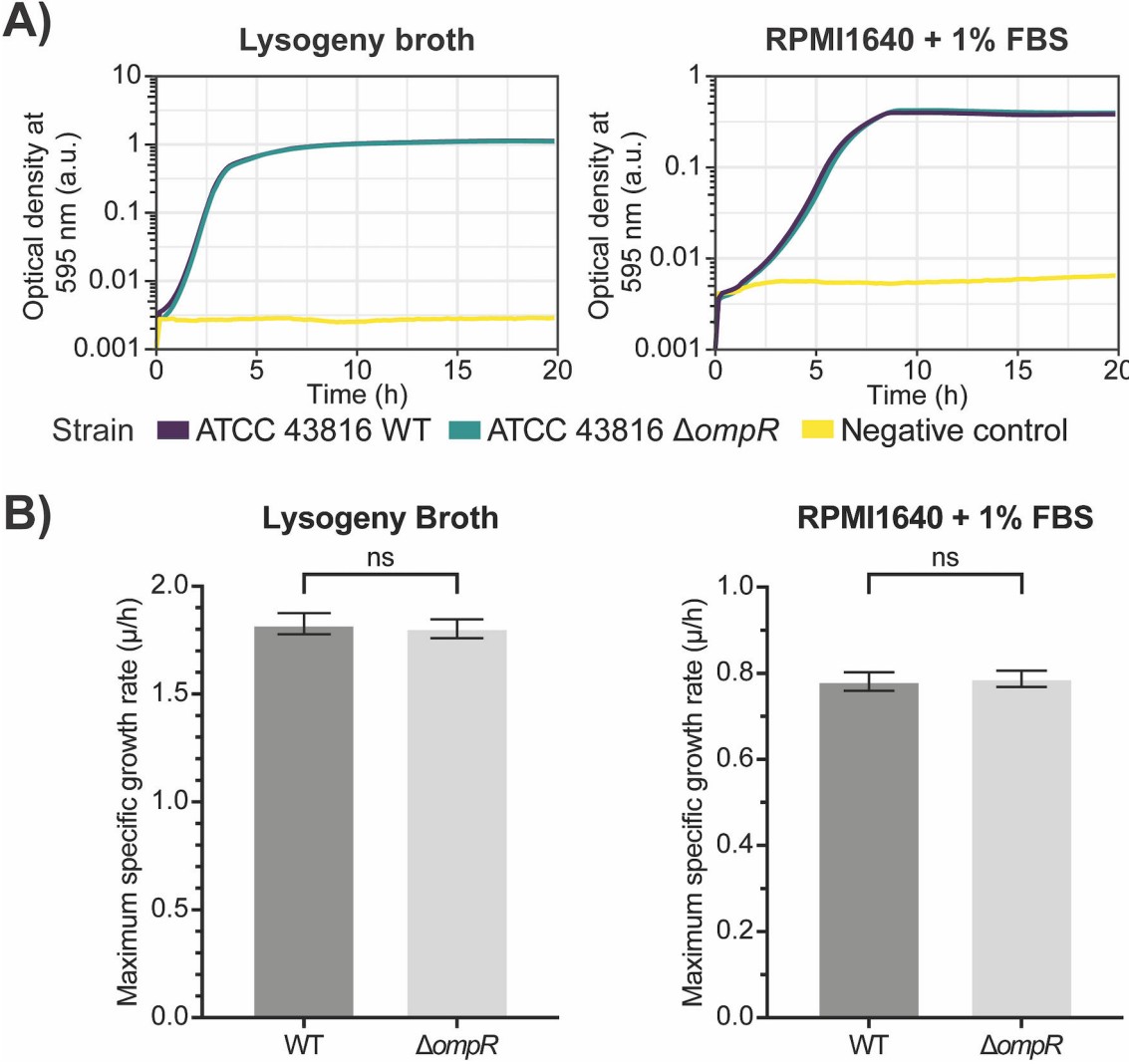

**FIG 1** Deletion of *ompR* in *K. pneumoniae* ATCC 43816 does not lead to a major fitness defect *in vitro*. (A) Determination of the growth kinetics of ATCC 43816 WT and ATCC 43816 Δ*ompR* showed that the deletion of *ompR* does not lead to changed growth kinetics in either LB or RPMI1640 complemented with 1% FBS. (B) Determination of the maximum growth speed as a measure of fitness showed that deletion of *ompR* does not lead to a reduced fitness in either LB or RPMI1640 complemented with 1% FBS. Statistical significance was determined using an unpaired *t*-test.

(WT) strain, in either LB or RPMI1640 supplemented with 1% FBS (Fig. 1B). Thus, a strain deficient in OmpR does not seem to have any growth defects *in vitro*.

In addition, we assessed whether the deletion of *ompR* in *K. pneumoniae* ATCC 43816 leads to changes in antibiotic susceptibility. We subjected the WT and the Δ*ompR* strain to standardized minimal inhibitory concentration determinations of several types of antibiotics. We observed that the susceptibility of the Δ*ompR* strain did not change more than twofold, compared with the WT strain, for all but the tetracycline molecules. The MIC values were fourfold higher for both tetracycline (1 µg/mL to 4 µg/mL) and minocycline (0.5 µg/mL to 2 µg/mL) for the Δ*ompR* strain compared with the WT strain (Table S1). These increases did not however make the Δ*ompR* strain phenotypically resistant to the tested tetracyclines, according to the breakpoints provided by the CLSI.

## Limited similarity in OmpR regulon between *K. pneumoniae* and other species during early host-pathogen interaction

Because OmpR is the transcriptional regulator part of the EnvZ/OmpR TCS, we questioned what consequences the *ompR* deletion has on gene expression of *K. pneumoniae* ATCC 43816. To investigate the gene expression, we utilized dual RNA-seq. The simultaneous genome-wide profiling of host and pathogen transcriptional responses using deep sequencing (i.e., dual RNA-seq) offers limited technical bias and higher efficiency compared with conventional approaches (e.g., single species approach, or array-based methods) (31).

As *K. pneumoniae* is a well-known lung pathogen, the pathogen was co-incubated with a monolayer of human A549 lung epithelial cells, which have been widely used as a model for respiratory infections (45, 46). The lung epithelial cells thus resemble a relevant substrate for the pathogen for the study of early host-pathogen interactions. We cultured the A549 lung epithelial cells without antibiotics to exclude any effects due to the residual presence of antibiotics or their downstream effects, for example, the presence of bacterial fragments (e.g., intracellular) and their effects on the host response. As the main interest in this experiment was to assess the early host-pathogen interaction, we selected the timepoint of 2 hours after the start of co-incubation (2 hpi) of the A549 and the *K. pneumoniae* cells as the endpoint of this infection model.

For the dual RNA-seq analysis, three samples of A549 lung epithelial cells infected with ATCC 43816 WT and two samples of A549 lung epithelial cells infected with the ATCC 43816 Δ*ompR* mutant strain were available (one Δ*ompR* replicate was unfortunately lost during sample preparation). Global analysis of the dual-RNA-seq libraries of early infection showed that an average of 54.7 million (range 36.2 million–74.0 million; Fig. S1A) reads mapped to the hybrid host-pathogen genome. Most aligned reads mapped onto the human genome (average 65.3%, range: 61.3%–68.9%; Fig. S1B). However, higher saturation and read coverage was observed for the bacterial genes (Fig. S1A). Normalized read counts of the RNA-seq data set showed that replicate infections with either WT or Δ*ompR* were reproducible (Pearson's $r > 0.995$; Fig. S1C) and that most of the transcriptional variation for either organism was attributable to the deletion of *ompR* (Fig. S1D). Successful depletion of rRNA was verified by counting the number of reads mapping to the features encoding rRNA (Table S2).

Within the pathogen data set, we observed that deletion of *ompR* in the genome of *K. pneumoniae* ATCC 43816 led to a statistically significantly $1.5 \times$ lower ($P_{adj} < 0.05$) expression of 12 genes, while 11 genes were statistically significantly $1.5 \times$ higher ($P_{adj} < 0.05$) expressed (Fig. 2; Table 1; Table S3). Genes encoding outer membrane porins *ompK35* and *ompC* (21) but also *dtpA* (20), encoding a di-/tripeptide permease, were lower expressed. We also observed higher expression of *mglB* (27), a galactose/glucose transporter. These genes can play a role in maintaining proper intracellular osmolarity in the bacterial cells by regulating transport of their specific substrates. We also observed a lower expression of *mrkA* (47), encoding a fimbrial type 3 subunit. Of the genes observed to be statistically significantly differentially expressed (Table 1), *dtpA*, *ilvC*, *mglB*, *ompC*, *ompF* (*ompK35* in *K. pneumoniae*), and *ompR* have been described to be regulated by

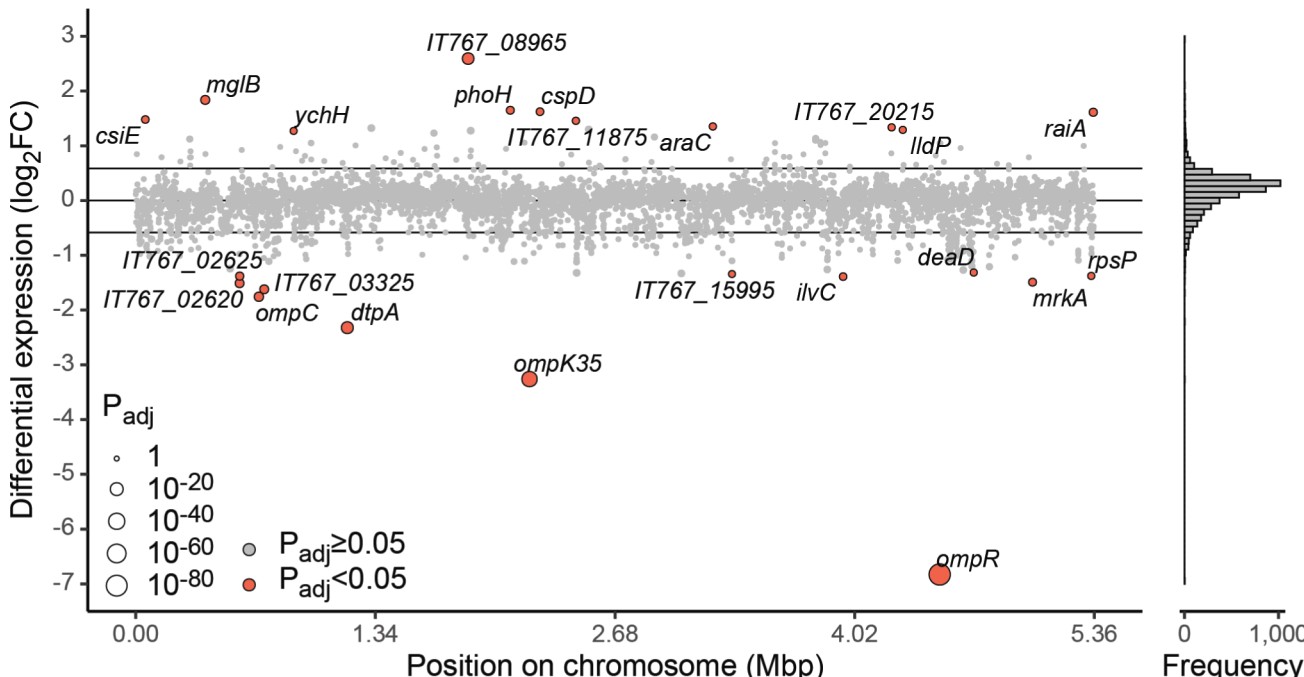

**FIG 2** OmpR-mediated differential expression in *K. pneumoniae* co-incubated with human epithelial cells. Fold changes in expression of *K. pneumoniae* ATCC 43816 genes in the ΔompR strain relative to WT on a log$_2$ scale. Point size indicates adjusted *P*-values on an inverse log$_{10}$ scale, and color indicates significance, compared with a 1.5-fold change in expression (horizontal lines). Gene names of significantly differentially expressed genes are indicated if annotated; locus tags are given otherwise. The histogram indicates the distribution of log$_2$FC values to illustrate point density.

OmpR in *E. coli* (13, 20, 21). In *Salmonella* species, *mglB*, *ompR*, and *ompC* (commonly known as *ompS1* in *Salmonella*) overlap with the genes observed in the OmpR regulon in *K. pneumoniae* (13, 27, 48).

Outside of the genes previously described to be in the OmpR regulon, we observed several other genes to be statistically significantly differentially expressed. Among others, IT767_03325, encoding a non-heme ferritin-like protein and a homolog of FtnB of *Salmonella* species, was lower expressed. In addition, we observed that IT767_02620 (previously renamed to *rmpC*), involved in the regulation of the biosynthesis of capsular polysaccharides (49), was lower expressed as well. Higher expression was observed for IT767_20215, predicted to encode a porin involved in glycerol uptake, and *lldP*, encoding a L-lactate permease. These last two could impact the maintenance of proper intracellular osmolarity.

In an attempt to identify the preferential DNA sequence bound by OmpR, we compared the sequences (500 bp) immediately preceding the genes that were differentially expressed in our setup. From this analysis, no conclusive consensus DNA sequence could be identified (data not shown).

## *K. pneumoniae* OmpR influences anti-bacterial responses by human lung epithelial cells

In addition to the pathogen response to deletion of *ompR* within the *in vitro* infection model, we analyzed the transcriptional response on the host side to study OmpR-specific host responses during interaction with *K. pneumoniae*. In the human data set, out of 57,816 annotated genes in the assembly of the human genome, 2,786 genes were filtered out because no reads had mapped to the feature (Table S4). A further 19,153 genes were filtered out due to a low normalized read count (Fig. S1A), leaving 35,877 genes (62%) for which differential expression was assessed (40).

**TABLE 1** Genes significantly differentially expressed in *K. pneumoniae* ΔompR compared with WT, when co-incubated with a monolayer of human A549 lung epithelial cells[a]

| Gene or locus tag | log$_2$FC | Adjusted *P*-value | (Predicted) function |
| --- | --- | --- | --- |
| *mpR* | −6.827565916 | 7.00509E−86 | Two-component system response regulator |
| *mpK35* | −3.259249606 | 1.68898E−34 | Porin |
| *dtpA* | −2.321517549 | 2.97161E−16 | Dipeptide/tripeptide permease |
| *mpC* | −1.757216339 | 0.000000947174 | Porin |
| IT767_03325 | −1.621397713 | 0.000015452404 | Non-heme ferritin-like protein |
| IT767_02620 | −1.511025745 | 0.000015452404 | DNA-binding response regulator |
| *mrkA* | −1.491525118 | 0.00176437549 | Type 3 fimbria major subunit |
| *ilvC* | −1.389780828 | 0.00546572241 | Ketol-acid reductoisomerase |
| IT767_02625 | −1.380606427 | 0.000503848469 | DMT family transporter |
| *rpsP* | −1.3772856 | 0.0263555292 | 30S ribosomal protein S16 |
| IT767_15995 | −1.342613157 | 0.0436857435 | tRNA-Leu |
| *deaD* | −1.31497228 | 0.0231368979 | ATP-dependent RNA helicase |
| *ychH* | 1.271180794 | 0.0253742485 | Stress-induced protein |
| *lldP* | 1.290650902 | 0.0264506555 | L-Lactate permease |
| IT767_20215 | 1.336512105 | 0.0170453364 | Aquaporin |
| *araC* | 1.352973572 | 0.0170453364 | Arabinose operon transcriptional regulator |
| IT767_11875 | 1.456031162 | 0.0180202346 | YbgS-like family protein |
| *csiE* | 1.479682873 | 0.00580507439 | Stationary phase-inducible protein |
| *raiA* | 1.611844528 | 0.000497823148 | Ribosome-associated translation inhibitor |
| *cspD* | 1.622883409 | 0.00278980858 | Cold shock-like protein |
| *phoH* | 1.648602795 | 0.00158781766 | Phosphate starvation-inducible protein |
| *mglB* | 1.837292397 | 0.000005778769 | Galactose/glucose ABC transporter substrate-binding protein |
| IT767_08965 | 2.591391864 | 1.07569E−14 | YmiA family putative membrane protein |

[a]Genes that are significantly differentially expressed (having an absolute log$_2$FoldChange (log$_2$FC) > log$_2$(1.5) and an adjusted *P* < 0.05) in the ΔompR strain compared with the WT strain. For each gene, the function, the adjusted *P*-value, and log$_2$FoldChange are given. For genes that have not been named, the predicted function (as per the published annotation) is given.

Of these 35,877 genes, we observed that 44 genes were significantly lower expressed (|log$_2$FC| < 0, $P_{adj}$ < 0.01) in the cells exposed to the ATCC 43816 Δ*ompR* bacteria, compared with the cells exposed to the WT bacteria. A large portion of these lower expressed genes encode non-coding RNAs, mostly small nucleolar RNAs but also small nuclear RNAs (Fig. 3A; Table S4). In addition to the non-coding RNAs, among the genes most negatively affected in expression in the mutant-exposed cells were multiple genes encoding ribosomal proteins (e.g., *RPSAP8* and *RPS27P13*) (Fig. 3A). In contrast, we found that no genes were significantly higher expressed (|log$_2$FC| > 0, $P_{adj}$ < 0.01) in the lung epithelial cells exposed to ATCC 43816 Δ*ompR* bacteria compared with the lung epithelial cells exposed to the WT bacteria (Fig. 3A; Table S4).

To better understand the patterns in the differentially expressed genes between the cells exposed to ATCC 43816 WT cells and those exposed to ATCC 43816 Δ*ompR* cells, we used a GO term enrichment analysis. By using a GO term enrichment analysis, the RNA-seq data can be assessed for an entire ontology (either biological processes, molecular functions, and/or cellular components), instead of on an individual gene level. When analyzing only those genes that were significantly differentially expressed, the GO term enrichment analysis confirmed that RNA processing pathways were significantly downregulated in lung epithelial cells exposed to ATCC 43816 Δ*ompR* cells (Table S5; Fig. S2).

However, as subtle (non-significant) changes in the expression of the genes that together constitute a biological process may sum up to significantly influence a process together, we also performed GO term gene set enrichment analysis using the log$_2$FC values of all 55,030 genes for which a fold change could be calculated. In addition to the significantly lower expression of RNA processing genes as identified before, this analysis also identified relative downregulation of gene sets related to household processes such as ribosomal functioning and respiration in the mutant-exposed cells (Fig. 3B; Table S6).

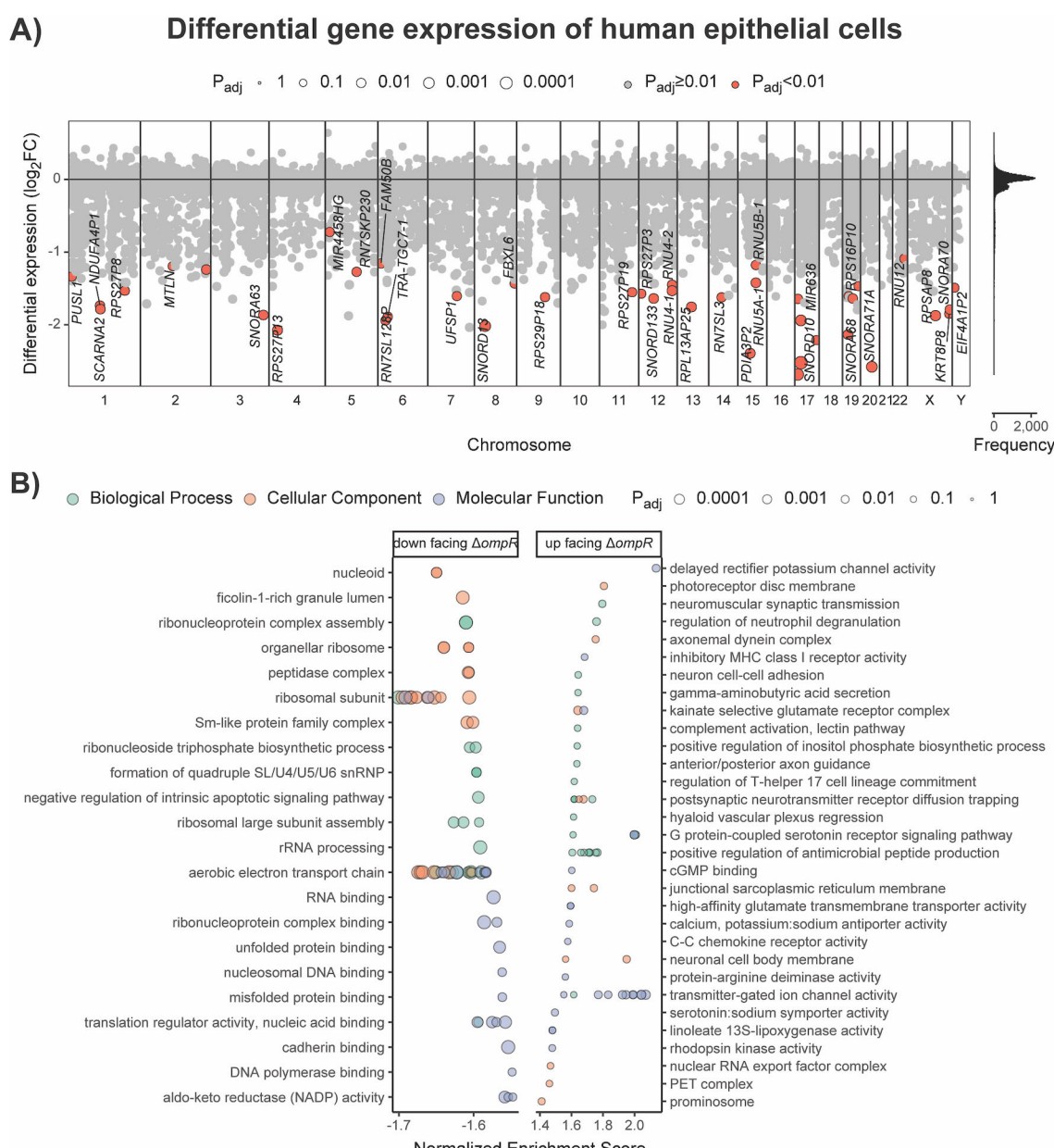

**FIG 3** Transcriptional response of human A549 lung epithelial cells in response to co-incubation with *K. pneumoniae* ATCC 43,816 WT, or ΔompR cells. (A) Differential expression over all loci per chromosome (1–22, X, Y) of cells challenged with the ΔompR strain relative to the WT strain. Significantly differentially expressed genes ($|\log_2FC| > 0$, $P_{adj} < 0.01$) are depicted in orange and with corresponding gene names if annotated. The histogram shows the distribution of the $\log_2FC$ values to indicate point density. (B) GO term gene set enrichment analysis (GSEA) of all $\log_2FC$ values. GO terms were clustered with the aPEAR package. Shown are the top 25 GO terms with the highest and lowest normalized enrichment scores with $P_{adj} < 0.1$ for each data base (Biological Process, Cellular Component, and Molecular Function).

In contrast, higher expression in the cells challenged with mutant bacteria was observed for the GO terms involved in antimicrobial peptide production, innate immune responses, and neuronal communication (Fig. 3B; Table S6).

Through dual RNA-seq in an *in vitro* host-pathogen model, we identified that deletion of *ompR* in *K. pneumoniae* ATCC 43816 leads to lower expression of several genes involved in biogenesis of membrane-related structures such as *ompC* (annotated as *ompS1* in *Salmonella* species) and *ompK35* but also of *mrkA* and IT767_02620 (*rmpC*). These genes, or the products that they regulate, have been previously described to be associated with virulence in diverse Enterobacteriaceae (5, 19, 47, 50–52). In addition, we

observed that the lung epithelial cells that were exposed to the mutant strain had higher expression of pathways involved in antimicrobial responses, such as processes involved in the complement system, and regulation of the production of antimicrobial peptides. These results, together with previous reports that OmpR is an important virulence factor in closely related bacterial species (18, 24, 29), suggest that the deletion of *ompR* in *K. pneumoniae* could also lead to a reduction in virulence.

### Deletion of *ompR* reduces virulence and infection of *K. pneumoniae* ATCC 43816

To assess the effects of the deletion of *ompR* on the virulence of *K. pneumoniae* ATCC 43816 and its ability to establish an infection in the lungs, male CD-1 mice were intranasally infected with exponential phase WT or $\Delta ompR$ bacteria.

The intranasal inoculation with $1.92 \times 10^6$ CFU of the WT strain resulted in mean bacterial lung titers of $1.59 \times 10^6$ and $9.75 \times 10^6$ CFU at 2 and 24 hpi, respectively (Fig. 4A), and in 100% mortality at 48 hpi (Fig. 4B). Inoculation with 1/10th of that dose resulted in mean bacterial titers of $8.97 \times 10^4$, $3.51 \times 10^7$, and $1.08 \times 10^8$ CFU per lung at 2, 24, and 48 hpi. In addition, decreased clinical severity scores and mortality at 48 hpi were observed compared with the higher inoculum (Fig. 4B). In contrast to inoculation with WT cells, intranasal inoculation with $1.59 \times 10^6$ CFU $\Delta ompR$ strain resulted in mean

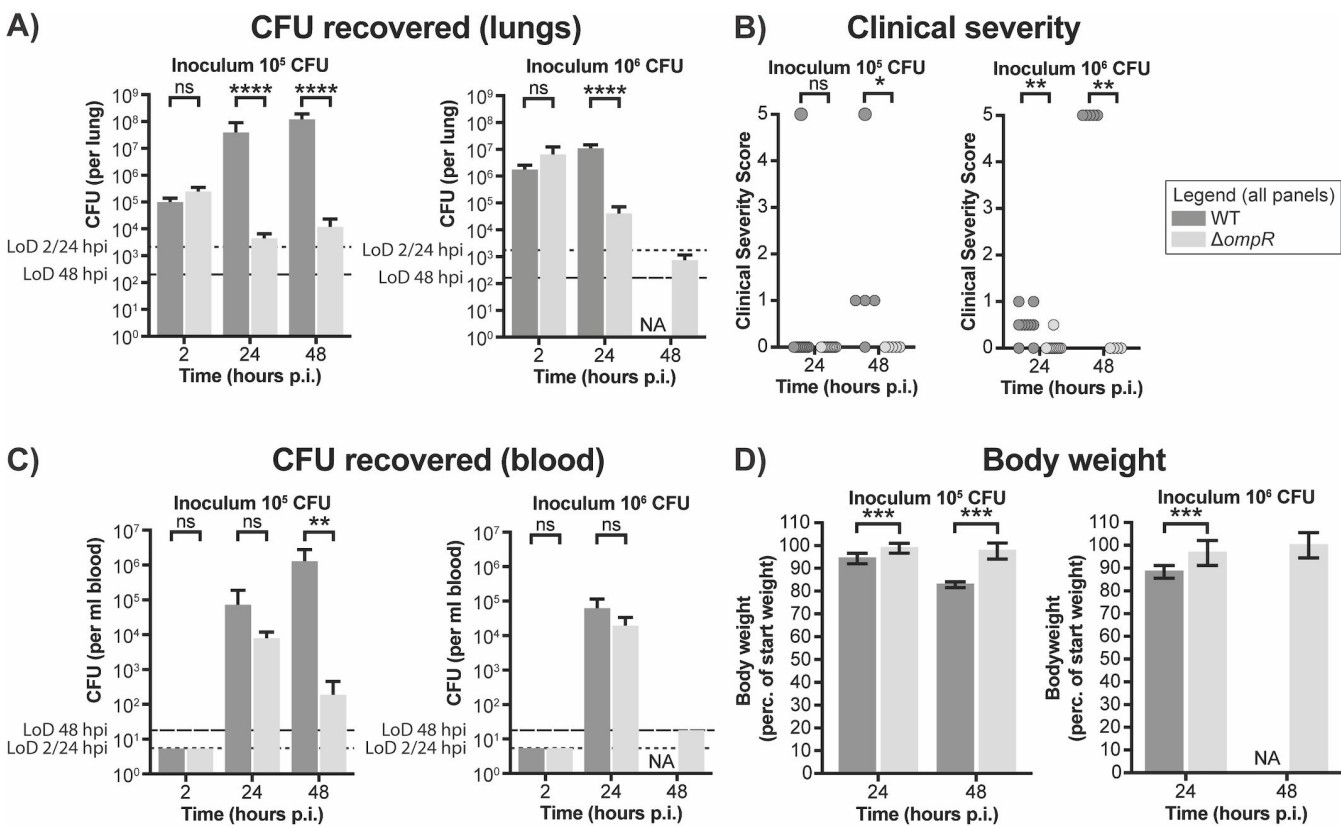

**FIG 4** Deletion of *ompR* in *K. pneumoniae* leads to decreased bacterial loads in a *K. pneumoniae* lung infection model. CD-1 mice (15 per group) were infected intranasally with *K. pneumoniae* ATCC 43816 WT or the isogenic ΔompR mutant strain. (A) The lungs were collected for bacterial titer determination. For the group infected with high inoculum of WT cells, no data were gathered at 48 hpi, because all mice of this group had died before reaching this timepoint. (B) Clinical severity scores were determined for each mouse at 2, 24, and 48 hpi. A score of 0 denotes no clinical symptoms, a score of 5 denotes death. (C) The blood was collected for bacterial titer determination. For the group infected with high inoculum of WT cells, no data were gathered at 48 hpi, because all mice of this group had died before reaching this timepoint. (D) The body weight of the infected mice was determined at 2, 24, and 48 hpi. Body weight was normalized to the weight at the beginning of the infection experiment. The legend is applicable to all panels. Statistical significance for the number of CFU recovered from the lungs and from the blood and for the body weight was determined using an unpaired *t*-test. Statistical significance for the clinical severity scores was determined using a ranked Mann-Whitney U-test. Dashed lines denote the limits of detection at 2/24 hpi (short dashes) and 48 hpi (long dashes).

bacterial lung titers of $5.80 \times 10^6$, $3.67 \times 10^4$, and $6.63 \times 10^2$ CFU at 2, 24, and 48 hpi, respectively (Fig. 4A). Inoculation with the lower dose of the Δ*ompR* strain resulted in mean bacterial lung titers of $2.22 \times 10^5$, $4.00 \times 10^3$, and $1.06 \times 10^4$ CFU at 2, 24, and 48 hpi, respectively. None of the animals infected with the Δ*ompR* strain died. Mice infected with the Δ*ompR* mutant strain had decreased clinical severity scores during the 48 hours of monitoring post-infection, compared with the mice infected with the WT strain (Fig. 4B).

At the 24-hpi timepoint, the lower number of CFUs recovered from the lungs of mice infected with the Δ*ompR* mutant strain compared with the WT strain did not reflect the number of cells recovered from the blood, where similar numbers of CFUs were recovered for both strains (Fig. 4C). In contrast, at the 48-hpi timepoint, more cells could be recovered from the blood of the mice inoculated with $10^5$ CFU of WT cells than that of the Δ*ompR* mutant strain. The deletion of *ompR* thus seemingly resulted in a decreased ability to establish lasting infections in the lungs and blood of the infected mice.

This decrease in virulence of the Δ*ompR* strain, as observed through the decreased number of CFUs recovered from the lungs and blood and the decreased clinical severity and mortality, was also reflected by a decreased loss of body weight in the infected mice. Mice infected with the Δ*ompR* mutant strain lost less body weight over the course of the infection experiment (Fig. 4D). These data suggest that the deletion of *ompR* results in a significant decrease in the virulence of *K. pneumoniae* ATCC 43816.

## DISCUSSION

In the present study, we set out to investigate the role of the EnvZ/OmpR TCS in gene expression, infection, and virulence of *K. pneumoniae* ATCC 43816. We observed that deletion of *ompR* in *K. pneumoniae* ATCC 43816 does not lead to an appreciable loss of fitness *in vitro*. In addition, susceptibility to clinically relevant antibiotics also did not change. Through dual RNA-seq, we showed changes in transcriptional activity of several genes, including virulence-associated genes (in *K. pneumoniae*) and antibacterial response genes (in human lung epithelial cells). The deletion of *ompR* heavily attenuated the virulence of *K. pneumoniae* ATCC 43816 in a murine infection model. These data suggest that OmpR is an important regulator of virulence and that it plays a crucial role in the ability of *K. pneumoniae* to establish invasive disease.

On the pathogen side, the deletion of *ompR* led to changes in expression of orthologous genes previously described to be controlled by OmpR in other species [including *dtpA*, *ompK35*, *ompC* (commonly known as *ompS1*), *mrkA*, and *mglB*] but also of genes outside of this previously established orthologous regulon. The gene commonly annotated as *ompC* in *E. coli* and *Salmonella* species was not significantly differentially expressed. The limited overlap between the OmpR regulons of *K. pneumoniae* and other Enterobacteriaceae is not unsurprising, as the OmpR regulons between *E. coli* and *Salmonella enterica* subspecies enterica serovar Typhimurium, a Gram-negative bacterium more closely related to *E. coli* than *K. pneumoniae*, also differ substantially (13, 17, 20–22, 26, 27). As we could not identify the preferential DNA sequence for the binding of OmpR by comparing the sequences preceding the genes that had been identified to be differentially expressed, future studies using ChIP-seq may elucidate the preferential DNA sequence bound by OmpR and its exact regulon in *K. pneumoniae*.

Through our dual-RNA-seq experiment, we observed downregulation of genes involved in virulence in the *ompR* deletion strain. Interestingly, we observe the downregulation of *rmpC* (annotated here as IT767_02620), which is involved in the regulation of capsule production in *K. pneumoniae*. This observation contrasts with earlier work that shows that deletion of *ompR* does not change the transcriptional activity of the *rmpADC* operon (30). Importantly, however, differences in experimental setup between our work (host-pathogen setup, cell culture medium) and the earlier performed work (Colombia blood agar plate) may explain these differences. We also identified that a homolog of ferritin-like protein FtnB (IT767_03325) was lower expressed, which has been described to be important in virulence in *Salmonella* (53). Lastly, the observed lower expression

of *mrkA*, encoding a fimbrial type 3 subunit, potentially results in the reduced presence of type 3 fimbriae. Together with the observed loss of hypermucoviscosity observed in previous work (30), these results could explain the attenuation in virulence of the mutant strain.

On the host side, no individual antibacterial response genes were significantly differentially expressed. However, GO term GSEA results do suggest that the epithelial cells mount an increased antibacterial response toward the mutant strain. This transcriptional response is constituted of increased antimicrobial peptide production, increased innate immune responses, and increased neuronal communication (54). The increased expression of these groups of genes suggests enhanced clearance and could in part explain the reduced virulence of the *ompR* knockout strain in the mouse infection model, in addition to the reported loss of hypermucoviscosity (30).

In our experiments, we confirm the previously reported reduced ability to establish an invasive infection of an OmpR-deficient ATCC 43816 strain (30). Interestingly however, the number of WT and Δ*ompR* cells that were recovered from the blood at 24 hpi was not significantly different, in contrast to the number of CFUs recovered from the lungs at that timepoint. This suggests that loss of OmpR does not affect the ability of *K. pneumoniae* to disseminate from the lungs to the blood. However, OmpR does seem crucial for *K. pneumoniae* for survival in both blood and lungs.

Although we have identified possible mechanisms through which a deletion of *ompR* may lead to reduced virulence *in vivo*, further experimentation may lead to a deeper biological understanding. The lack of a clear transcriptional signature in the A549 cells may be the result of the fact that only two samples were available for the *ompR* deletion mutant, impacting statistical power. In addition, an overwhelming transcriptional response due to the presence of lipopolysaccharides, which are considered the prototypical pathogen-associated molecular pattern, might obscure any subtle host transcriptional effects from other missing epitopes as the result of the loss of OmpR. Any further experimentation should also include a complementation mutant, which we lack in this work. It would also be interesting to compare uninfected A549 cells with the transcriptome of cells infected with *K. pneumoniae*. In addition, the use of more sophisticated *in vitro* model systems such as cell cultures with an air-liquid interface may further improve the understanding of the biological function of OmpR in infection. In any case, the current experimental results show that *ompR* does not seem crucial during the early stages of lung infection but is rather required for survival at later stages.

Because of the central role of TCSs in bacterial adaptation to specific environmental cues, the development of inhibitory drugs against these systems has been explored before (9, 29). Our work and work performed by others previously suggest that, because of the key role of EnvZ/OmpR in pathogenesis, newly developed anti-infective compounds targeting EnvZ/OmpR could be potent drugs to treat infections. These potential drugs would have a low potential for drug resistance development since there is no impact on bacterial fitness. Although a decrease in susceptibility to tetracyclines was noted, tetracyclines are rarely used clinically to treat infections with *K. pneumoniae*. The deletion of *ompR* did not change the susceptibility to antibiotics typically used to treat infections with *K. pneumoniae* in clinical cases, including cephalosporins and carbapenems. The development of anti-virulence drugs targeting TCSs might thus be a fruitful approach to treat infections with multidrug-resistant bacteria.

## ACKNOWLEDGMENTS

We thank V. Benes (GeneCore, European Molecular Biology Laboratory, Heidelberg) for his support in library preparation and sequencing. Additionally, we would like to thank C.R. Aranzamendi Esteban and S. El Aidy for their access to human cell culture laboratory. We thank L. Ferrari and A. Felici at Aptuit (today Evotec) for the planning, conducting, and analysis of the mouse experiments.

A.B.J. is supported through a Postdoctoral Fellowship grant (TMPFP3_210202) by the Swiss National Science Foundation (SNSF). Work in the lab of J.-W.V. is supported

by the SNSF (project grants 310030_192517 and 310030_200792) and the SNSF NCCR "AntiResist" program (51NF40_180541). The funders had no role in study design, data collection and analysis, decision to publish, or preparation of the manuscript.

A.B.J. conceived, designed, and performed experiments, analyzed the data, and wrote the first draft of the manuscript. V.d.B. analyzed data and edited the manuscript. R.A. conceived, designed, and performed experiments and analyzed data. V.T. conceived, designed, and performed experiments, analyzed data, and edited the manuscript. C.K. conceived and designed experiments and supervised the study. M.P. conceived and designed experiments, supervised the study, and edited the manuscript. J.-W.V. conceived and designed experiment, supervised the study, and edited the manuscript.

All authors reviewed and approved the final version of the manuscript.

## AUTHOR AFFILIATIONS

[1]Department of Fundamental Microbiology, Faculty of Biology and Medicine, University of Lausanne, Lausanne, Switzerland
[2]Molecular Genetics Group, University of Groningen, Groningen Biomolecular Sciences and Biotechnology Institute, Centre for Synthetic Biology, Groningen, the Netherlands
[3]BioVersys AG, Basel, Switzerland

## AUTHOR ORCIDs

Axel B. Janssen http://orcid.org/0000-0002-9865-447X
Vincent de Bakker http://orcid.org/0000-0003-1019-3558
Rieza Aprianto http://orcid.org/0000-0003-2479-7352
Vincent Trebosc http://orcid.org/0000-0002-8961-1371
Michel Pieren http://orcid.org/0000-0001-7896-254X
Jan-Willem Veening http://orcid.org/0000-0002-3162-6634

## FUNDING

| Funder | Grant(s) | Author(s) |
|---|---|---|
| Swiss National Science Foundation | TMPFP3_210202 | Axel B. Janssen |
| Swiss National Science Foundation | 310030_192517 | Jan-Willem Veening |
| Swiss National Science Foundation | 310030_200792 | Jan-Willem Veening |
| Swiss National Science Foundation | 51NF40_180541 | Jan-Willem Veening |

## AUTHOR CONTRIBUTIONS

Axel B. Janssen, Conceptualization, Formal analysis, Investigation, Methodology, Visualization, Writing – original draft | Vincent de Bakker, Data curation, Formal analysis, Investigation, Methodology, Visualization, Writing – review and editing | Rieza Aprianto, Conceptualization, Data curation, Formal analysis, Investigation, Methodology, Writing – review and editing | Vincent Trebosc, Conceptualization, Formal analysis, Methodology, Writing – review and editing | Christian Kemmer, Conceptualization, Supervision, Writing – review and editing | Michel Pieren, Conceptualization, Supervision, Writing – review and editing | Jan-Willem Veening, Conceptualization, Funding acquisition, Project administration, Supervision, Writing – review and editing

## DATA AVAILABILITY

The raw Illumina sequencing reads of the transcriptomics data set are available in Gene Expression Omnibus (GEO) under accession number GSE123964.

## ETHICS APPROVAL

Animal experiments were performed by Aptuit (today Evotec; Verona, Italy). All experiments involving animals were carried out in accordance with directive 2010/63/EU of the European Union governing the welfare and protection of animals, implemented by Italian Legislative Decree number 26/2014, and in accordance with Aptuit's policy on the care and use of laboratories animals. All animal studies were reviewed by an Animal Welfare Body (Aptuit) and approved by Italian Ministry of Health.

## ADDITIONAL FILES

The following material is available online.

### Supplemental Material

**Figure S1 (Spectrum03966-23-s0001.tiff).** Composition and mapping of dual RNA-seq Illumina sequencing reads.
**Figure S2 (Spectrum03966-23-s0002.tiff).** GO term enrichment analysis of significantly downregulated genes in lung epithelial cells in response to ΔompR K. pneumoniae.
**Supplemental legends (Spectrum03966-23-s0003.docx).** Legends for supplemental Figures S1 and S2.
**Table S1 (Spectrum03966-23-s0004.xlsx).** MIC values for ATCC 43816 WT and ATCC 43816 ΔompR.
**Table S2 (Spectrum03966-23-s0005.xlsx).** Read counts for the features encoding the ribosomal RNAs in the host and K. pneumoniae ATCC 43816 genomes.
**Table S3 (Spectrum03966-23-s0006.xlsx).** DESeq2 analysis of the genes in the K. pneumoniae genome.
**Table S4 (Spectrum03966-23-s0007.xlsx).** Results of DESeq2 analysis of the genes in the human genome.
**Table S5 (Spectrum03966-23-s0008.xlsx).** GO term enrichment analysis for differentially expressed genes in lung epithelial cells.
**Table S6 (Spectrum03966-23-s0009.xlsx).** Full GO term gene set enrichment analysis for human lung epithelial cells.

### Open Peer Review

**PEER REVIEW HISTORY (review-history.pdf).** An accounting of the reviewer comments and feedback.

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
