## [Reviewer comments · Microbiology Spectrum]

Microbiology Spectrum

Klebsiella pneumoniae OmpR facilitates lung infection through transcriptional regulation of key virulence factors

Axel Janssen, Vincent de Bakker, Rieza Aprianto, Vincent Trebosc, Christian Kemmer, Michel Pieren, and Jan-Willem Veening

Corresponding Author(s): Jan-Willem Veening, Universite de Lausanne

Review Timeline:

Submission Date:

November 20, 2023

Accepted:

November 22, 2023

Editor: Christopher LaRock

Reviewer(s): The reviewers have opted to remain anonymous.

Transaction Report:

DOI: <https://doi.org/10.1128/spectrum.03966-23>

Re: Spectrum03966-23 (Klebsiella pneumoniae OmpR facilitates lung infection through transcriptional regulation of key virulence factors)

Dear Prof. Jan-Willem Veening:

Your manuscript has been accepted, and I am forwarding it to the ASM production staff for publication. Your paper will first be checked to make sure all elements meet the technical requirements. ASM staff will contact you if anything needs to be revised before copyediting and production can begin. Otherwise, you will be notified when your proofs are ready to be viewed.

Sincerely,
Christopher LaRock
Editor
Microbiology Spectrum